# WHEN SPIKING NEURAL NETWORKS MEET TEMPORAL ATTENTION IMAGE DECODING AND ADAPTIVE SPIKING NEURON

**Xuerui Qiu, Zheng Luan, Zhaorui Wang, Rui-Jie Zhu** [*]
University of Electronic Science and Technology of China
{sherry.qiu, zheng.luan, zhaorui_wang, ridger}@std.uestc.edu.cn

## ABSTRACT

Spiking Neural Networks (SNNs) are capable of encoding and processing temporal information in a biologically plausible way. However, most existing SNN-based methods for image tasks do not fully exploit this feature. Moreover, they often overlook the role of adaptive threshold in spiking neurons, which can enhance their dynamic behavior and learning ability. To address these issues, we propose a novel method for image decoding based on temporal attention (TAID) and an adaptive Leaky-Integrate-and-Fire (ALIF) neuron model. Our method leverages the temporal information of SNN outputs to generate high-quality images that surpass the state-of-the-art (SOTA) in terms of Inception score, Fréchet Inception Distance, and Fréchet Autoencoder Distance. Furthermore, our ALIF neuron model achieves remarkable classification accuracy on MNIST (99.78%) and CIFAR-10 (93.89%) datasets, demonstrating the effectiveness of learning adaptive thresholds for spiking neurons. The code is available at this link

## 1 INTRODUCTION

Spiking neural networks (SNNs) are a promising technique for image processing tasks, as they offer significant computational advantages over conventional neural networks Qiu et al. (2023). Recent advances in SNNs have enabled image classification Fang et al. (2021) and generation Kamata et al. (2022) using backpropagation Wu et al. (2018). However, existing methods have some limitations. For instance, the fully spiking variational autoencoder (FSVAE) proposed by Kamata et al. (2022) uses fixed weights and does not exploit the rich temporal information of SNN outputs. Moreover, the parametric leaky integrate-and-fire (LIF) neuron model developed by Fang et al. (2021) only considers frequency adaptation and neglects threshold dynamics.

## 2 METHOD

### 2.1 ADAPTIVE LEAKY-INTEGRATE-AND-FIRE MODEL (ALIF)

Among all popular spiking neuron models, LIF model has significantly reduced computational demand and is commonly recognized as the simplest model while retaining biological interpretability Burkitt (2006). LIF model is characterized by the following differential equation:

$$\tau \frac{du(t)}{dt} = -u(t) + x(t) \tag{1}$$

where $x(t)$ represents the input from the presynaptic spiking neurons, $u(t)$ represents the membrane potential of the neuron at time step $t$, and $\tau$ is a constant number. Additionally, spikes will fire if $u(t)$ exceeds the threshold $V_{th}$. To understand the working mechanism in detail, LIF is expressed more iteratively Wu et al. (2018) in **Appendix A.1**.

For ALIF, the coefficients $\tau$ and $V_{th}$ are learnable. Moreover, we discuss the initialization of $\tau$ and $V_{th}$ in **Appendix Tab 5** . By understanding the iterative expression of LIF neurons, its backpropagation process is shown in **Appendix A.1** following chain rules.

### 2.2 TEMPORAL ATTENTION IMAGE DECODING (TAID)

Here, we propose a TAID mechanism to build temporal-wise links between different output spiking sequences and transform sequences into images. The output tensor of the networks is $S \in$

---

[*]Corresponding Author

$\mathbf{R}^{T \times C \times H \times W}$, where $T$ is the simulation step and $C$ is the channel size. Then, TAID squeezes the output $S$ and gets the mean matrix $X$. Its element $X_u$ can be described as below:

$$X_u = \frac{1}{H \times W \times C} \sum_{H}^{i=1} \sum_{W}^{j=1} \sum_{C}^{k=1} S_{i,j,k}^u \tag{2}$$

After the squeeze operation, a learnable matrix $W$ is used to connect the temporal-channel-wise relationships with the mean matrix. Then putting the element-wise product weight matrix $W$ and mean matrix $X$ in the Sigmoid function, we finally get the fusion matrix $F$. More precisely, its element $F_i$ is:

$$F_i = \sigma(\sum_{T}^{i=1} W_i X_i) \tag{3}$$

Then, we employ the $F$ to measure the potential correlation between the temporal and channel on the network output tensor $S$. In the end, by calculating the mean of operated $S$ in the temporal dimension, we can get the image.

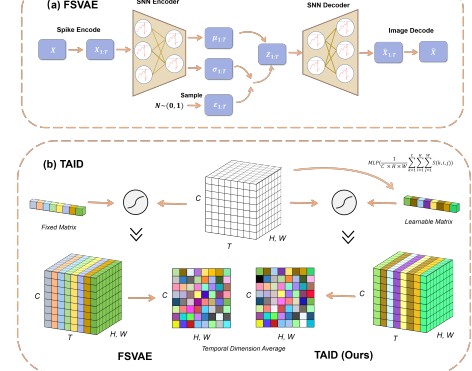

Figure 1: (a): the workflow of fully spiking variation autoencoder. (b): comparison between the original image decoding method and TAID.

## 3 EXPERMENTS

### 3.1 IMAGE GENERATION

In order to make better use of temporal dimension information for spike to image decoding on image generation, we build a fully spiking variational autoencoder (FSVAE) by our TAID. Details of the network, training, and metrics can be seen in the **Appendix A.2 and A.3**. The workflow chart of this FSVAE with TAID applied on image decoding is shown in Fig. **1**. As a comparison method, we prepared vanilla VAEs of the same network architecture built with ANN and trained on the same settings. Moreover, we get a SOTA result on image generation. Table 1-2 demonstrates that for all datasets, our TAID outperforms FSVAE Kamata et al. (2022) and ANN in all metrics, for CelebA and CIFAR10 datasets.

Table 1: Results for CIFAR10 on generation.

| Method | IS ↑ | FID ↓ | FAD ↓ |
|---|---|---|---|
| ANN | 2.59 | 229.6 | 196.9 |
| Kamata et al. (2022) | 2.94 | 175.5 | 133.9 |
| This work (TAID) | **3.53** | **171.1** | **120.5** |

Table 2: Results for CelebA on generation.

| Method | IS ↑ | FID ↓ | FAD ↓ |
|---|---|---|---|
| ANN | 3.23 | 92.53 | 156.9 |
| Kamata et al. (2022) | 3.69 | 101.6 | 112.9 |
| This work (TAID) | **4.31** | **99.54** | **105.3** |

Table 3: Results for MNIST on classification.

| Method | $T$ | Accuracy |
|---|---|---|
| Fang et al. (2020) | 10 | 99.46 |
| Fang et al. (2021) | 8 | 99.72 |
| This work (ALIF) | 8 | **99.78** |

Table 4: Results for CIFAR10 on classification.

| Method | $T$ | Accuracy |
|---|---|---|
| Rathi & Roy (2020) | 10 | 92.64 |
| Fang et al. (2021) | 8 | 93.72 |
| This work (ALIF) | 8 | **93.89** |

### 3.2 IMAGE CLASSIFICATION

Using our ALIF, we get a competitive result on image classification. Details of the network and training can be seen in the **Appendix A.2 and A.3**, and the result is shown in Table 3 and Table 4. On the static dataset MNIST, our method can achieve the highest classification accuracy with only 8 time step. On the static dataset CIFAR10, we obtain the competitive result over the prior method with binary spikes.

## 4 CONCLUSION

In this paper, we propose a novel Temporal Attention Image Decoding (TAID) method and Adaptive LIF (ALIF) neuron. TAID can make full use of the temporal information to perform better in generating images. Moreover, ALIF focuses on the threshold of the neuron and outperforms Top 1 Accuracy (99.78%) on MNIST. The results of extensive experiments support our proposed novel method.

## URM STATEMENT

The authors acknowledge that all authors of this work meets the URM criteria of ICLR 2023 Tiny Papers Track.

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

## A APPENDIX

### A.1 CHAIN RULE OF ALIF

For better computational tractability, the ALIF model can be described as an explicitly iterative version.

$$x_{t+1,n}^i = \Sigma_j w_n^j o_{t+1,n-1}^j \tag{4}$$

$$u_{t+1,n}^i = \tau u_{t,n}^i (1 - o_{t,n}^i) + x_{t+1,n}^i \tag{5}$$

$$o_{t+1,n}^i = h(u_{t+1,n}^i - V_{th}) \tag{6}$$

Here, $t$ and $n$ respectively represent the indices of the time step and $n$-th layer, and $o^j$ is its binary output of $j$-th neuron. Furthermore, $w^j$ is the synaptic weight from $j$-th neuron to $i$-th neuron and by altering the way that $w^j$ is linked, we can implement convolutional layers, fully connected layers, etc. Since $h(\cdot)$ represents the Heaviside function and Eq. 6 is non-differentiable. The following derivatives of the surrogate function can be used for approximation.

$$\frac{\partial o^i_{t+1,n}}{\partial u^i_{t+1,n}} = \frac{1}{a}\operatorname{sign}\left(\left|u^i_{t+1,n} - V_{\text{th}}\right| < \frac{a}{2}\right) \tag{7}$$

With this expressly ALIF neuron, backpropagation of the $\tau$ and $V_{th}$ process can be finished by the next chain rule.

$$\frac{\partial L}{\partial \tau} = \frac{\partial L}{\partial o^i_{t+1,n}}\frac{\partial o^i_{t+1,n}}{\partial u^i_{t+1,n}}\sum_{m=0}^{t-1}\left(\frac{\partial u^i_{t+1-m,n}}{\partial \tau}\cdot\prod_{j=t+2-m}^{t+1}\frac{\partial u^i_{j,n}}{\partial u^i_{j-1,n}}\right)$$

$$=\frac{\partial L}{\partial o^i_{t+1,n}}\frac{1}{a}\operatorname{sign}\left(\left|u^i_{t+1,n} - V_{\text{th}}\right| < \frac{a}{2}\right)\sum_{m=0}^{t}\left(u^n_{t-i}\left(1 - o^i_{t-m,n}\right)\prod_{j=t+2-m}^{t+1}\tau\left(1 - o^i_{j-1,n}\right)\right) \tag{8}$$

$$\frac{\partial L}{\partial V_{th}} = \frac{\partial L}{\partial o^i_{t+1,n}}\frac{\partial o^i_{t+1,n}}{\partial V_{th}} = -\frac{\partial L}{\partial o^i_{t+1,n}}\frac{1}{a}\operatorname{sign}\left(\left|u^i_{t+1,n} - V_{\text{th}}\right| < \frac{a}{2}\right) \tag{9}$$

Table 5: Ablation study of $\tau$ and $V_{th}$ on MNIST

| Learnable | Init $\tau$ | Init $V_{th}$ | Accuracy |
|:---:|:---:|:---:|:---:|
| ✗ | 0.25 | 0.2 | 99.64 |
| ✗ | 0.5 | 0.2 | 99.52 |
| ✗ | 0.5 | 0.5 | 99.41 |
| ✓ | 0.25 | 0.2 | **99.78** |
| ✓ | 0.5 | 0.2 | 99.71 |
| ✓ | 0.5 | 0.5 | 99.62 |

## A.2 NETWORK STRUCTURE DETAILS

**Image generation:** In order to verify the effectiveness of the ALIF neuron and TAID and compare with the best performance counterpart method fairly, FSVAE Kamata et al. (2022), we follow the network architecture of FSVAE settings, which is shown in Kamata et al. (2022). Moreover, the architecture of ANN VAE is the same as FSVAE.
**Image classification:** In order to verify the effectiveness of the ALIF neuron and compare it with the best performance counterpart method fairly, PLIF Fang et al. (2021), we follow the network architecture of PLIF settings.

Table 6: The network architecture setting for each dataset. cxkysz and **MPkysz** is the **Conv2D** and **MaxPooling** layer with output channels = x, kernel size = y and stride = z. **DP** is the spiking dropout layer. **FC** denotes the fully connected layer. **AP** is the global average pooling layer. the **\*n** indicates the structure repeating n times

| Datasets | Network Architecture |
|:---:|:---:|
| MNIST | {c128k3s1-BN-ALIF-MPk2s2}*2- DP-FC2048-ALIF-DP-FC100-ALIF-APk10s10 |
| CIFAR-10 | {{c256k3s1-BN-ALIF}*3- MPk2s2}*2-DP-FC2048-ALIF- DP-FC100-ALIF-APk10s10 |

A.3 TRAINING DETAILS:

**Image generation:** We employ the AdamW optimizer Loshchilov & Hutter (2017), which trains 300 epochs at 0.001 learning rate and 0.001 weight decay. The batch size is set to 200. Moreover, the time step is set to 16. For CIFAR10 Krizhevsky et al. (2009), we used 50,000 images for training and 10,000 images for evaluation. For CelebA Liu et al. (2015), we used 162,770 images for training and 19,962 images for evaluation. The input images were scaled to $64 \times 64$. And, we apply log-likelihood evidence lower bound (ELBO) as the loss function:

$$
\begin{aligned}
ELBO =& \mathbb{E}_{q(\boldsymbol{z}_{1:T}|\boldsymbol{x}_{1:T})} \left[ \log p\left( \boldsymbol{x}_{1:T} \mid \boldsymbol{z}_{1:T} \right) \right] \\
& - \mathrm{KL} \left[ q\left( \boldsymbol{z}_{1:T} \mid \boldsymbol{x}_{1:T} \right) \| p\left( \boldsymbol{z}_{1:T} \right) \right],
\end{aligned}
\tag{10}
$$

where the first term is the reconstruction loss between the original input and the reconstructed one, which is the mean square error (MSE) in this model. The second term is the Kullback-Leibler (KL) divergence, representing the closeness of prior and posterior.

Moreover we use Inception Score (IS) Heusel et al. (2017) , Fréchet inception distance (FID) Salimans et al. (2016) and Fréchet Autoencoder distance (FAD) Kamata et al. (2022) as evaluation-metrics. Moreover, FAD Kamata et al. (2022) is trained by an autoencoder on each dataset before-hand. And it is used to measure the Fréchet distance of the autoencoder's latent variables between sampled and real images. We sampled 5,000 images to measure the distance. As a comparison method, we prepared vanilla VAEs of the same network architecture built with ANN and trained on the same settings.

**Image classification:** We employ the Adam optimizer , which trains 1000 epochs. The batch size is set to 16. All datasets are the same as the above. Moreover, our loss function is same as the Fang et al. (2021)

A.4 GENERATED IMAGES:

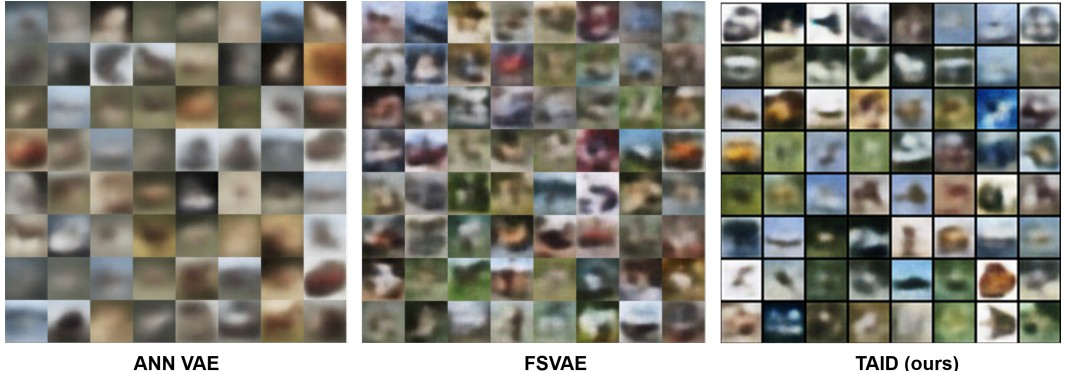

ANN VAE   FSVAE   TAID (ours)

Figure 2: Generated images of CIFAR10

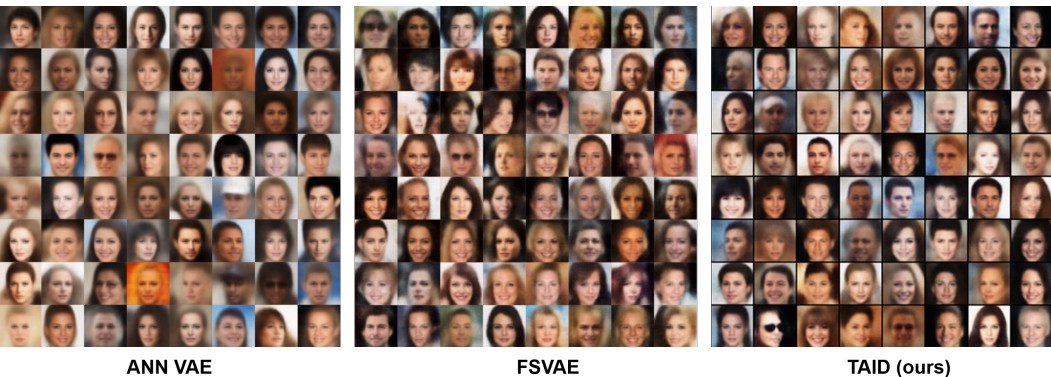

ANN VAE   FSVAE   TAID (ours)

Figure 3: Generated images of CelebA

