# OpenReview forum: "When Spiking Neural Networks Meet Temporal Attention Image Decoding and Adaptive Spiking Neuron"
_ICLR.cc/2023/TinyPapers — Submitted to Tiny Papers @ ICLR 2023_

### Official Review · Reviewer_psKB · 2023-03-20

**Confidence:** 3

**Summary Of Contributions:**

This submission proposes an spiking variational autoencoder built out of adaptive LIF neurons, and shows that it outperforms previously-proposed spiking architectures for image generation.

**Rating:**

Clear, Correct, and Reproducible (CCR): a submission which meets the reviewing criteria

**Strengths And Weaknesses:**

I assess the paper's strengths and weaknesses with respect to each of the reviewing criteria below:

- **Clarity**: I found the description of the method and of the results fairly easy to follow.

- **Correctness**: The experimental evaluation appears solid, and the claimed improvements on past work seem to be adequately demonstrated.

- **Reproducibility**: The architecture and experimental setup are described in sufficient detail to allow reproduction. It could be useful to release code upon acceptance.

- **Follows basic requirements**: The paper fits within the page limit, and is adequately anonymized.

**Suggested Changes:**

1. As mentioned above, please consider releasing code to allow easy reproduction of your results.

2. A small formatting suggestion to improve readability: use \citep or \parencite rather than \citet or \textcite when appropriate. For instance, in the first sentence, set "Roy et al. (2019)" as "(Roy et al. 2019)".

3. There are a few places where I think some editing could improve the clarity of the manuscript. For example, what do the authors mean when they write "LIF is expressed more iteratively?" Based on referring to Appendix 1, they appear to use "iterative" to refer to the first-order discretization of the LIF dynamics; this could be stated more clearly.

4. The diagram in Figure 1 is so small as to be nearly illegible. I'd suggest that the authors make it larger, and, if necessitated by length constraints, move it to the Appendix.

5. The non-spiking ANN baseline in Tables 1-4 does not seem to be clearly described. What activation function is used? How far is this network from the the state-of-the-art?

---

> ### Author Response · Authors · 2023-04-10
> **Reply**
>
> Thank you for your great recommendation. We will carefully revise vague descriptions in the paper. And we will open the source code as soon as possible.

---

### Official Review · Reviewer_7TLW · 2023-04-01

**Confidence:** 4

**Summary Of Contributions:**

This work proposes a method that combines temporal attention with an adaptive Leaky-Integrate-and-Fire neuron model to improve image decoding in Spiking Neural Networks (SNNs). The proposed method achieves state-of-the-art results in MNIST and CIFAR-10 benchmarks.

**Rating:**

High Potential (HP): a submission which meets the reviewing criteria and has potential to make an impact on the field

**Strengths And Weaknesses:**

### Strengths:
1. The paper is well-motivated and clear to understand.
2. The authors have conducted thorough experimentation on MNIST and CIFAR-10 to justify the correctness of their approach. I especially like the ablation studies in Table 5 which help in further understanding of the proposed design choices.

### Weakness:
1. Regarding reproducibility, the authors have not provided any code nor they have expressed any intention of open sourcing the code in the submission. I request the authors to clarify on this.


**Suggested Changes:**

- I recommend an acceptance of this submission with a High Potential rating. While the method does borrow design choices from prior methods (Kamata et al. 2022 and Fang et al. 2021), the overall method performs very well on the selected benchmarks.

- I would also ask the authors to discuss more on Table 5. Also a small paragraph on how tau and Vth are tuned would help in further understanding of the approach.

- **Change** : In Table 2, it looks like ANN has the lowest FDD score. Therefore, its value should be in bold font and not your method's FDD score.

---

> ### Author Response · Authors · 2023-04-10
> **Reply**
>
> Thank you for your great recommendation. We will open the source code as soon as possible.

---

### Author Response · Authors · 2023-05-30
**Author Decision**

We wish to opt-in for archival.

---

### Comment · Area_Chair_RzU5 · 2023-06-02
**ICLR Tiny Paper Archival**

This work meets the threshold for archival, contents the URM statement and is deanonymized.

---

### Meta-Review · Area_Chair_RzU5 · 2023-04-09

**Recommendation:** Invite to present
**Confidence:** 5

**Metareview:**

This paper proposes two new methods named TAID (Temporal Attention Image Decoding) and ALIF (Adaptive LIF) to improve image decoding for Spiking Neural Networks. The resulting findings show that the new methods consistently outperform the state-of-the-art in several utility metrics. Overall, **the paper is clear and well-written**. **Conclusions are justified by the findings of experiments**.

**Summary:**

The paper proposes new methods to improve image decoding for Spiking Neural Networks, which consistently outperforms the state-of-the-art in several utility metrics for image generation and classification accuracy. Reviewers remarked that ANN in Table 2 has a lower FID metric and should be in bold instead of the proposed TAID. Reviewers also recommended to please open-source the code for ease of reproducibility.

**Comments And Feedback To The Authors:**

Table 1 could be improved by adding "This work (TAID)" and, similarly, Table 2 with "This work (ALIF)".

**Reason For Not Giving A Higher Recommendation:**

Reproducibility was not possible to evaluate and reviewers recommend open-sourcing the codes.


**Reason For Not Giving A Lower Recommendation:**

The proposed methods are promising and have the potential to be impactful for image generation/decoding using Spiking Neural Networks.

---

> ### Author Response · Authors · 2023-04-10
> **Reply**
>
> Thank you for your great recommendation. We will carefully revise vague descriptions in the paper. And we will open the source code as soon as possible.Our code is now  available at https://github.com/bollossom/ICLR_TINY_SNN

---

### Decision · Program_Chairs · 2023-04-10

Invite to present